# Assessing the Response of Diverse Sesame Genotypes to Waterlogging Durations at Different Plant Growth Stages

**DOI:** 10.3390/plants10112294

**Published:** 2021-10-25

**Authors:** Mohammad Habibullah, Shahnaz Sarkar, Mohammad Mahbub Islam, Kamal Uddin Ahmed, Md. Zillur Rahman, Mohamed F. Awad, Abdelaleim I. ElSayed, Elsayed Mansour, Md. Sazzad Hossain

**Affiliations:** 1Department of Agricultural Botany, Sher-e-Bangla Agricultural University, Sher-e-Bangla Nagar, Dhaka 1207, Bangladesh; habibagriculg@gmail.com (M.H.); shahnazsarkar2@hotmail.com (S.S.); mahbubislam_sau@yahoo.com (M.M.I.); kuahamed@yahoo.com (K.U.A.); 2Syngenta Bangladesh Limited, Green Rowshan Ara Tower (5th & 6th Floor), 55 Satmasjid Road, Dhanmondi, Dhaka 1205, Bangladesh; 3Department of Agronomy and Haor Agriculture, Sylhet Agricultural University, Sylhet 3100, Bangladesh; zillur.agronomy@sau.ac.bd; 4Department of Biology, College of Science, Taif University, P.O. Box 11099, Taif 21944, Saudi Arabia; m.fadl@tu.edu.sa; 5Department of Biochemistry, Faculty of Agriculture, Zagazig University, Zagazig 44511, Egypt; 6Department of Crop Science, Faculty of Agriculture, Zagazig University, Zagazig 44519, Egypt

**Keywords:** sesame, genotypes, waterlogging, tolerance, growth, yield, principal component analysis

## Abstract

Sesame is sensitive to waterlogging, and its growth is devastatingly impacted under excess moisture conditions. Thus, waterlogging tolerance is crucial to alleviate yield constraints, particularly under expected climate change. In this study, 119 diverse sesame genotypes were screened for their tolerance to 12, 24, 48, and 72 h of waterlogging relative to non-waterlogged conditions. All plants died under 72 h of waterlogging, while 13.45%, 31.93%, and 45.38% of genotypes survived at 48, 24, and 12 h, respectively. Based on the seedling parameters and waterlogging tolerance coefficients, genotypes BD-7008 and BD-6985 exhibited the highest tolerance to waterlogging, while BD-6996 and JP-01811 were the most sensitive ones. The responses of these four genotypes to waterlogged conditions were assessed at different plant growth stages—30, 40, and 50 days after sowing (DAS)—versus normal conditions. Waterlogging, particularly when it occurred within 30 DAS, destructively affected the physiological and morphological characteristics, which was reflected in the growth and yield attributes. Genotype BD-7008, followed by BD-6985, exhibited the highest chlorophyll and proline contents as well as enzymatic antioxidant activities, including superoxide dismutase (SOD), peroxidase (POD), and catalase (CAT). These biochemical and physiological adjustments ameliorated the adverse effects of waterlogging, resulting in higher yields for both genotypes. Conversely, JP-01811 presented the lowest chlorophyll and proline contents as well as enzymatic antioxidant activities, resulting in the poorest growth and seed yield.

## 1. Introduction

Sesame (*Sesamum indicum* L.) is one of the oldest and most essential oilseed crops worldwide [1]. Its total cultivated area is nearly 13 million hectares, which produce about 6.5 million tons annually [2]. Globally, it is utilized in several edible uses, cosmetics preparations, drugs, paints, perfumes, soaps, lubricants, fungicides, and insecticides [3]. Sesame is cultivated in different soil types, but it greatly thrives in well-drained fertile soils with a neutral pH [4]. However, it is commonly grown in rainfed conditions and is often exposed to flooded conditions [5]. As a result, sesame is tremendously sensitive to waterlogging and continuous heavy rains compared to other crops [1,6]. Even brief periods of excess moisture result in considerable reductions in seed yield [7]. Waterlogging causes a variety of morphological, physiological, and biochemical alterations that adversely affect plant growth, development, and production [8,9,10].

Waterlogging stress occurs when the soil pores become fully saturated, preventing normal air circulation in the rhizosphere [8,11]. With this switch from aerobic to anaerobic respiration, the reduction in O_2_ availability reduces the energy generated by the plant. In addition, oxygen deficiency (hypoxia) or its complete absence (anoxia) in the soil causes a reorganization of metabolic fluxes, resulting in an immediate energy crisis and plant death [12]. With a shortfall in oxygen availability, the processes that depend on oxygen, such as carbon assimilation and photosynthate utilization, are suppressed [13,14]. Moreover, oxygen depletion and rising carbon dioxide concentrations devastatingly impact root growth and disturb the functional relationships between roots and shoots [15,16,17,18]. Additionally, in the waterlogged soil, compounds such as ethylene, manganese, carbon dioxide, and iron concentrations may rise, causing chronic toxicity as well as anaerobic microbial metabolites, which may accumulate [19]. Furthermore, low-oxygen stress leads to increasing toxic reactive oxygen species (ROS), such as hydroxyl radical (^•^OH), singlet oxygen (^1^O_2_), hydrogen peroxide (H_2_O_2_), and superoxide radical (O^2•−^). Accumulation of ROS leads to lipid peroxidation, protein degradation, and enzyme inactivation. Furthermore, biomolecules, such as nucleic acids, and cellular structures, such as membranes, can be damaged by ROS production [20,21].

Plants have a complex defense antioxidant system to mitigate the oxidative deteriorations caused by ROS [22,23,24]. The antioxidant system includes low molecular weight antioxidants, such as proline and antioxidant enzymes, which include ascorbate peroxidase (APX), catalase (CAT), superoxide dismutase (SOD), glutathione reductase (GR), and guaiacol peroxidase (GPX) [25,26,27,28]. Waterlogging tolerance has been related to the ability of the plants to scavenge ROS and reduce harmful effects, with considerable differences among the genotypes [29,30,31,32].

Waterlogging stress may worsen due to expected changes in the climate, which may increase the frequency of excess moisture and continuous heavy rains [33]. Consequently, developing tolerant genotypes is the most suitable and cost-effective approach to cope with this situation [34,35,36]. However, a deeper understanding of the underlying tolerant mechanism is necessary to improve waterlogging tolerance in sesame. Therefore, it is crucial to assess diverse sesame genotypes under waterlogging stress to identify tolerant accessions and investigate the tolerance mechanisms. Moreover, the duration of waterlogging and the occurrence of excess moisture during the growth stage have different impacts on physiological metabolism and, accordingly, plant growth and productivity [26,37]. Studies on the response of sesame genotypes under different durations of waterlogging at diverse growth stages are lacking. Hence, the current study was designed (i) to evaluate diverse sesame genotypes under various durations of waterlogging and identify valuable tolerant genetic resources; and (ii) investigate the response of sesame genotypes to waterlogged conditions at different plant growth stages and identify the critical stage that is most sensitive to waterlogging.

## 2. Materials and Methods

### 2.1. Laboratory Experiment

A total of 119 genotypes were assessed for their adaptive capacity under waterlogged conditions (Table 1). Seeds of different genotypes were obtained from the Plant Genetic Resource Center (PGRC), Bangladesh Agricultural Research Institute (BARI), Joydebpur, Gazipur, Bangladesh. First, 13 (4–4–4–1) healthy seeds of each genotype were placed in a Petri dish lined with moist filter paper. The seeds were left in an incubator (EYELA LTI-700, Japan) for three days under controlled conditions (temperature of 25 ± 2 °C; RH of 65–70%) for uniform germination. After three days, the Petri dishes were transferred into a control growth room at a temperature of 25 ± 2 °C with 60% relative humidity and 8 and 16 h of light and dark. Then, 10 mL of distilled water with a 1000-fold diluted Hyponex solution (Type: 5-10-5, Hyponex, Yodogawa-ku, Osaka, Japan) was applied to each Petri dish. After that, 5 mL of the Hyponex solution was added each day to every Petri dish to keep the seeds moist and compensate for evaporation losses. After five days, uniform seedlings were selected and divided into five groups. Then, five uniform seedlings were selected and treated for 12, 24, 48, and 72 h using 200 mL of distilled water to avoid oxygen exchange with the air (water stress), and a control group was treated with a standard tap water supply to keep the seeds moist. The experiment was laid out in a completely randomized design (CRD) with five treatments and replicated three times.

The seedling lengths (cm) and fresh weights (g) were measured for each genotype under the control and waterlogged conditions after 8 days. In addition, the waterlogging tolerance coefficient (WTC) was recorded for each genotype in percentage using the following formula:(1)$$WTC\;\left(weight\right)=\frac{Seedling\;fresh\;weight\;under\;waterlogged\;conditions\;}{Seedling\;fresh\;weight\;under\;control\;conditions}\times 100$$
(2)$$WTC\;\left(length\right)=\frac{Seedling\;length\;under\;waterlogged\;conditions\;}{Seedling\;length\;under\;control\;conditions}\times 100$$

Moreover, growth inhibition was estimated as a percentage relative to the length of the shoots and rootlets from the control (tap water).

### 2.2. Pot Experiment

#### 2.2.1. Experimental Design and Treatments

The experiment was conducted using a completely randomized design (CRD) with five replications at the Bangladesh Agricultural Research Institute (BARI), Joydebpur, Gazipur (23°59′ N, 90°24′ E). Four genotypes were used in this experiment based on the obtained results of the preliminary laboratory experiment. The sesame genotypes used were divided into two tolerant (BD-7008 and BD-6985) and two sensitive (JP-01811 and BD-6996). Plastic pots that were 35 cm deep and 25 cm in diameter were filled with 10 kg of sandy loam soil mixed with coir pith compost (Table 2). Five seeds were sown in each pot, and after complete emergence, one plant was maintained per pot and the excess plants were thinned out. The genotypes were evaluated under waterlogged conditions at three different growth stages—30, 40, and 50 days after sowing (DAS) in comparison to the control (normal conditions, without waterlogging). The duration of waterlogging was 48 h and the water levels were maintained at 3 cm above the soil surface by replenishing frequently. After the treatment period, the water was drained out from the pots and the plants grew to maturity.

#### 2.2.2. Assessment of Chlorophyll and Proline Content

The chlorophyll contents of the leaf samples were estimated following the procedure described by Arnon [38] and expressed as mg g^−1^ of fresh weight. Weighed quantities of leaf samples (0.5 g) were collected from the third fully expanded leaves and cut into small pieces. These pieces were put into test tubes and incubated overnight at room temperature with 10 mL of DMSO and an 80% acetone mixture (1:1 *v/v*). The colored solution was transferred into a measuring cylinder, and the absorbance was measured at 663 nm and 645 nm.

Proline colorimetric determination proceeded according to the method by Bates et al. [39], based on proline’s reaction with ninhydrin. Fresh leaf tissue (0.5 g) was homogenized in 10 mL of 3% sulfosalicylic acid in ice. The homogenate was centrifuged at 11,500× *g* for 15 min. Then, 2 mL of the filtrate was mixed with 2 mL of acid ninhydrin and 2 mL of glacial acetic acid. After incubation at 100 °C for 1 h, the mixture was cooled, and 4 mL of toluene was added. The optical density of the chromophore containing toluene was read spectrophotometrically at 520 nm, using toluene as a blank. The amount of proline was estimated by comparison with a standard curve.

#### 2.2.3. Assessment of Enzymatic Antioxidants Activity

Superoxide dismutase (SOD) activity (EC 1.15.1.1) was estimated based on the xanthine–xanthine oxidase system according to the method by Hossain et al. [40]. The reaction mixture contained 50 mM of a KP buffer, 2.24 mM of nitro blue tetrazolium (NBT), 0.1 units of catalase, 2.36 mM of xanthine, and 0.1 unit of xanthine oxidase. Then, the change in the absorbance of the solution was recorded at 560 nm for 1 min, and the activity of SOD was expressed as unit mg^−1^ protein (the amount of enzyme required to inhibit NBT reduction by 50%). Peroxidase (POD) activity (EC: 1.11.1.7) was assessed following the method of Hemeda and Klein [41]. It was determined spectrophotometrically at 25 °C at 470 nm using guaiaco1 and H_2_O_2_ as the hydrogen donor and substrate. The substrate mixture contained 10 mL 1% guaiacol, 10 mL 0.3% hydrogen peroxide and 100 mL 0.05 M sodium phosphate buffer (pH 6.5). Catalase (CAT) activity (EC: 1.11.1.6) was determined following the procedure of Csiszár et al. [42] by monitoring the decline in absorbance at 240 nm for 1 min, and due to the degradation of H_2_O_2_, an extinction coefficient of 39.4 M^−1^ cm^−1^ was used. The reaction mixture contained 50 mM KP buffer (pH 7.0), 15 mM H_2_O_2_, and an enzyme extract (the final volume was 700 µL).

#### 2.2.4. Morphological and Yield Contributing Attributes

The morphological characters were measured after 92 days from sowing. Plant height (cm), the number of leaves per plant, the shoot fresh and dry weights (g), and the root fresh and dry weights (g) were recorded. The yield-contributing characteristics assessed at harvest (92 DAS) included the number of branches per plant, the number of pods per plant, the number of seeds per pod, the thousand seed weight (g), and seed yield (g/plant).

### 2.3. Statistical Analysis

The data for all the growth and physiological parameters were subjected to ANOVA, appropriate for a completely randomized design (CRD). Differences among treatments for all measurements were compared using LSD, and values were reported as significant at *p*-values of < 0.01. All analyses were applied using R statistical software version 4.4.1.

## 3. Results

### 3.1. Laboratory Experiment

#### 3.1.1. Survival Percentage

At the seedling stage, 119 diverse sesame genotypes were grown under four durations of waterlogged conditions (12, 24, 48, and 72 h) to identify tolerant and sensitive genotypes. All genotypes died after 72 h of waterlogged conditions; however, 16, 38, and 54 genotypes survived after 48, 24, and 12 h of waterlogged conditions, respectively (Table 3). The mortality percentages varied from 54.2% to 100% under 12 to 72 h of waterlogging (Figure 1).

The survival percentages were calculated for sixteen genotypes that survived under 12, 24, and 48 h of waterlogged conditions, and they are presented in Table 4. The survival percentages varied from 40% (JP-01811 and BD-6996) to 100% (BD-6985) under 48 h of water stress. The maximum survival rate was recorded for genotypes BD-6985 and BD-7008 (100%), and the lowest survival rate was recorded for JP-01811 (40%) under 48 h of water stress.

#### 3.1.2. Seedling Measurements

The genotypes BARI-Til-4, BD-6959, GP-83-3, JP-01811, BD-7018, and BD-7004 exhibited the tallest lengths under control conditions (Figure 2A). In addition, BD-7018, BD-7008, BD-6998, JP-14003, JP-03013, BD-7004, BD-6985, and JP-00411 displayed the highest seedling length values after 48 h of waterlogging. On the contrary, the lowest seedling length values were recorded for BD-6996, BD-6991, and GP-83-3. Genotypes BD-6996, BD-6985, GP-5, BD-7018, JP-14003, and JP-03013 presented the highest values of fresh seedling weight under control conditions (Figure 2B), while BD-6985, BD-7004, BD-6998, BD-7018, JP-14003, JP-03013, JP-00411, and BD-7008 presented the highest fresh weights under waterlogging at 48 h. BD-6996, JP-01811, and BD-6991 exhibited the lowest fresh weights.

#### 3.1.3. Waterlogging Tolerance and Growth Inhibition

The waterlogging tolerance coefficients and growth inhibition rates were estimated for the sixteen sesame genotypes at the seedling stage under 48 h of waterlogging (Table 5). The genotypes responded differently to waterlogging stress. Genotypes BD-7008 and BD-6985 demonstrated the highest tolerance coefficient values and lowest inhibition percentages. On the contrary, BD-6996 and JP-01811 presented the lowest tolerance coefficient values and the highest inhibition percentages. Accordingly, these four genotypes were selected to be evaluated in the pot experiment.

### 3.2. Pot Experiment

#### 3.2.1. Physiological Response to Waterlogged Conditions

Waterlogging immensely reduced the chlorophyll content in all evaluated genotypes (Figure 3A). Genotypes BD-7008 and BD-6985 exhibited the highest chlorophyll contents under normal and waterlogged conditions, while JP-01811 and BD-6996 displayed the lowest values (Figure 3A). Additionally, the proline content increased under waterlogging compared to normal conditions (Figure 3B). Genotype BD-7008, followed by BD-6985, exhibited the highest values of proline content under waterlogged conditions, while the lowest values were presented by JP-01811 and BD-6996.

Enzymatic antioxidant (SOD, POD, and CAT) activities increased in all genotypes under waterlogged conditions compared to normal conditions (Figure 4). Genotype BD-7008 exhibited the highest activities of SOD, POD, and CAT under waterlogged conditions, followed by BD-6985, while the lowest values were exhibited by JP-01811 and BD-6996. Genotype BD-7008 displayed 41.7, 42.5, and 29.9% increases in SOD, POD, and CAT, respectively, under waterlogged conditions compared to control conditions. Similarly, BD-6985 exhibited increases in SOD, POD, and CAT by 28.4, 31.3, and 25.8%, respectively, under waterlogged conditions compared to normal conditions. In contrast, JP-01811 presented the lowest increases in SOD, POD, and CA, with values of 17.2, 9.4, and 6.9% under waterlogged conditions, compared to normal conditions.

#### 3.2.2. Morphological Characters

The evaluated genotypes exhibited significant differences in plant height, the number of leaves per plant, the number of branches per plant, shoot fresh weight, shoot dry weight, root fresh weight, and root dry weight (Table 6). Genotypes BD-6985 and BD-7008 presented the highest values of all the aforementioned characteristics (Table 6). In contrast, JP-01811 and BD-6996 displayed the lowest values of all growth parameters. The main effect of BD-6985 surpassed BD-6996 in all the above-mentioned characteristics by 44.4, 28.6, 28.9, 17.6, 16.3, 19.5, and 31.7%, respectively. Waterlogging stress, at all investigated growth stages, significantly decreased the growth parameters compared to normal conditions. However, waterlogging at 30 DAS steeply and significantly decreased all morphological characters compared to the other stages (Table 6).

Conversely, the least hazardous stage was recorded for 40 DAS, compared to 30 and 50 DAS. Waterlogging at 30 DAS decreased the plant height, the number of leaves per plant, the number of branches per plant, shoot fresh weight, shoot dry weight, fresh root weight, and root dry weight by 13.0, 28.9, 40.9, 27.7, 21.0, 18.6, and 23.7%, respectively, in comparison to normal conditions. In addition, waterlogging at 40 DAS reduced the characteristics by 3.4, 10.6, 17.2, 9.9, 13.0, 11.4, and 7.9%, respectively, compared to normal conditions. Moreover, waterlogging at 50 DAS reduced the aforementioned characteristics by 8.2, 16.9, 31.1, 18.5, 21.3, 5.7, and 10.5%, respectively, compared to control conditions. The growth parameters significantly differed among the genotypes with each waterlogging treatment (Table 6). Waterlogging at all investigated growth stages substantially decreased the growth parameters compared to normal conditions. Similar to the results of the main effects, genotypes BD-6985 and BD-7008 exceeded BD-01811 and BD-6996 in all growth parameters under all waterlogging treatments. Additionally, the interaction results ascertained that the 30 DAS growth stage was the most sensitive growth stage for waterlogging stress for all the evaluated genotypes. The lowest values of all growth parameters occurred with waterlogging at 30 DAS for all genotypes compared to the other treatments (Table 6). However, the 40 DAS growth stage was more tolerant than 30 and 50 DAS.

#### 3.2.3. Yield Traits

The genotypes exhibited substantial differences in the number of pods per plant, number of seeds per pod, 1000-seed weight, days to fruit set, days to maturity, and seed yield per plant (Table 7). Genotypes BD-6985 and BD-7008 possessed the highest values of all the above-mentioned measured yield traits (Table 7). In contrast, JP-01811 and BD-6996 showed the lowest values of all yield traits. The waterlogging treatments drastically decreased the yield traits compared to control conditions. Notably, waterlogging at 30 DAS severely and significantly reduced all yield traits compared to the other growth stages (Table 7). Waterlogging at 30 DAS reduced the number of pods per plant, number of seeds per pod, 1000-seed weight, days to fruit set, days to maturity, and seed yield per plant by 8.1, 8.1, 30.3, 6.8, 5.5, and 32.7%, respectively, compared to normal conditions. The negative impact of waterlogging at 40 DAS was the lowest compared to 30 and 50 DAS; it reduced yield traits by 2.7, 3.9, 17.2, 3.1, 2.4, and 18.8, respectively, compared to the normal conditions. The interaction results prove that the yield traits of genotypes BD-6985 and BD-7008 under all waterlogging treatments were superior to BD-01811 and BD-6996. In addition, 30 DAS was the most sensitive growth stage for waterlogging stress, followed by 50 and 40 DAS in all the evaluated genotypes.

#### 3.2.4. Interrelationships among the Evaluated Traits and Treatments

Principal component analysis (PCA) was applied in order to study the relationships among the evaluated traits and treatments, as displayed in Figure 5. The first two PCAs explained 91.59% of the variability. PCA1 accounted for 78.10% of the variation and appears to be associated with the waterlogging treatments. Regardless of the genotypes, the normal conditions (Cont) are situated on the positive side of PCA1, followed by waterlogging at 40 and 50 DAS and, finally, waterlogging at 30 DAS on the end of the negative side of PCA1. PCA2 accounted for 13.49% of the variation and appears to be related to sesame genotypes in the following order: BD-7008, BD-6985, JP-01811, and BD-6996. PCA2 divided the genotypes into two groups: the tolerant ones, BD-7008 and BD-6985, are situated on the positive side of the axis, and the sensitive ones, BD-6996 and JP-01811, are located on the negative side of the axis. Notably, the sensitive genotypes, BD-6996 and JP-01811, experienced minor effects, as described by the slight distances in the multi-dimensional space compared to the tolerant ones, BD-7008 and BD-6985, which are more scattered, inferring dissimilarity (Figure 5). As presented previously, the seed yield and its attributes are positively associated with the normal conditions (along the positive side of the PCA1 axis) and negatively associated with waterlogging at 30 DAS, which is further along the PCA2 axis. The adjacent vectors of seed yield and all its attributes reflect a positive association. Indeed, seed yield was strongly associated with the 1000-seed weight, the number of branches per plant, shoot fresh weight, shoot dry weight, and the number of leaves per plant.

## 4. Discussion

Waterlogging is one of the major abiotic stresses that cause biochemical, physiological, anatomical, and morphological alterations that translate into severe crop yield losses [43]. It initiates hypoxia (low oxygen) or anoxia (absence of oxygen), which leads to direct root damage and indirect leaf wilting under transient or sustained flooded conditions [44]. Climate change is expected to bring no rainfall for long periods, followed by heavy and prolonged precipitation, causing devastating floods [45,46]. Subsequently, field crops will face flooded conditions, leading to a drastic global reduction in food production. Accordingly, developing flood-tolerant genotypes has become urgent in order to cope with these constraints. Sesame is very sensitive to waterlogging; nevertheless, there are considerable differences among its genotypes [6,7]. Therefore, it is necessary to screen diverse sesame genetic resources under different waterlogging durations to assess their levels of flooding tolerance. In the present study, 119 diverse sesame genotypes were evaluated at the seedling stage under four waterlogging durations versus normal, non-waterlogged conditions. Seedling tests under controlled conditions are commonly carried out to screen for waterlogging tolerance [47]. Genetic variability for flooding tolerance can be assessed indirectly by survival percentage and crop damage indices [48]. The survival percentage was estimated, and it substantially varied between waterlogging durations. After 72 h of waterlogging, all the genotypes died, while the maximum survival (45.38%) was recorded at 12 h. Prolonged periods of waterlogging cause the complete absence of oxygen (anoxia), which inhibits the ability of roots to supply water and nutrients, hinders respiration and photosynthesis, and causes root rot and death [49,50,51,52]. Accordingly, all genotypes died after 72 h of waterlogging, while 16 survived after 48 h, 38 after 24 h, and 54 after 12 h of waterlogged conditions.

Waterlogging tolerance coefficients (WTCs) and seedling parameters were employed to distinguish the tolerant and sensitive genotypes out of the sixteen genotypes that survived under 12, 24, and 48 h of waterlogging. WTCs and seedling parameters can provide a reliable preliminary picture of the tolerance and sensitivity of sesame genotypes to flooding. Higher values expressed a higher tolerance to waterlogged conditions compared to lower ones. There was considerable variation among the sixteen sesame genotypes. Based on the WTCs and seedling parameters, genotypes BD-7008 and BD-6985 exhibited the highest tolerance to waterlogging, while BD-6996 and JP-01811 were the most sensitive ones. A pot experiment was applied to further investigate the response of tolerant and susceptible genotypes to waterlogging at different growth stages of 30, 40, and 50 DAS versus normal, non-waterlogged conditions. The degree of adverse impacts of waterlogged soils depends on the plant growth stage, the duration of flooding, soil type, growth conditions, and genotypes [44,53]. The maximum growth and yield traits were obtained under the control condition (non-waterlogged), while the physiological, developmental, and agronomic traits were destructively affected by flooding for 48 h, particularly at the early growth stage (30 DAS). Immediately after waterlogging, plants cannot follow any adaptive mechanisms, which results in the extreme unfitness of plants at the early stage of growth [54,55,56]. This is probably the reason that 30 DAS sesame exposed to waterlogging showed pronounced effects in all genotypes.

Under waterlogged conditions, the scavenging abilities of reactive oxygen species (ROS) of sesame normally decreased. The activities of enzymatic antioxidants have a crucial role in plant survival during prolonged flooding stress and preventing cell damage [57,58]. Therefore, the effective antioxidant system is positively and significantly associated with waterlogging tolerance. In the present study, the activities of the SOD, POD, CAT, and proline content increased in the tolerant sesame genotypes (BD-7008 and BD-6985) under waterlogged conditions compared to the sensitive ones (BD-6996 and JP 01811). Thus, the enzymic antioxidants provided a better balance between the ROS generated by waterlogging and detoxification, which enhanced the plant’s ability to cope with flooding stress [59]. These results are in accordance with the previous findings of Sairam et al. [30], Damanik et al. [60], and Xu et al. [61], since they demonstrated higher activities of enzymatic antioxidants in the waterlogging-tolerant genotypes than the sensitive ones. Moreover, Xu et al. [61] depicted an increased proline content under flooding stress in the tolerant sesame genotypes compared to the sensitive ones. Proline acts as a vital osmolyte for osmotic adjustment and contributes to stabilizing cell structures and protecting membranes and proteins against ROS [62]. Therefore, the enzymatic antioxidants activities and proline content could be employed as indirect indices to screen sesame genotypes for waterlogging tolerance. Furthermore, the stimulating effects of antioxidants enzyme activities and proline attributed positively to ameliorating waterlogging tolerance and, hence, exhibited higher growth and productivity. Accordingly, the tolerant sesame genotypes exhibited significantly higher growth and yield traits than the sensitive ones. The results of the pot experiment confirm the results of the seedling trial regarding the classification of the sesame genotypes based on their tolerance to waterlogging. Therefore, the seedling parameters and waterlogging tolerance coefficients (WTCs) at the seedling stage are efficient and reliable measurements for distinguishing tolerant and sensitive genotypes to waterlogging stress.

The PC biplot is an efficient statistical analysis to visualize the interrelationships among the evaluated treatments and traits [63,64,65,66]. The present results indicate that waterlogging at 30 DAS was located on the extreme negative side of the PCA1 axis compared to the normal, non-waterlogged conditions on the positive side, while 40 and 50 DAS were intermediate. This corroborates that the most critical period for waterlogging stress is the early growth stage, at 30 DAS, compared to the other growth stages at 40 and 50 DAS. Moreover, the PC biplot displays the evaluated sesame genotypes in different patterns: the tolerant ones were situated on the positive side of the PCA2 axis, while the sensitive ones were located on the negative side. Furthermore, the PC biplot demonstrates the reduction in growth and seed yield traits due to waterlogging stress, as they are opposite to waterlogging at 30 DAS and are positively associated with the normal conditions. Seed yield had a strong association with the 1000-seed weight and the number of branches per plant, which could be employed as rapid and indirect indicators to seed yield under waterlogging stress.

## 5. Conclusions

The evaluated sesame genotypes possess distinct genetic diversity, which is reflected in their ability to survive under waterlogged conditions. Genotypes BD-7008 and BD-6985 are considered to be highly tolerant of waterlogging stress, while BD-6996 and JP-01811 are the most sensitive. The tolerant genotypes exhibited the highest activities of enzymatic antioxidants—SOD, POD, and CAT—and proline content, compared to the sensitive ones. Accordingly, these antioxidant enzymes and proline content could be indirect indices to screen sesame genotypes for waterlogging tolerance. The physiological, developmental, and agronomic traits were destructively affected by flooding, particularly at the early growth stage, at 30 DAS, compared to 40 and 50 DAS. Accordingly, the growth stage is critical to waterlogging stress. The results of the seedling experiment are consistent with the pot trial, based on identifying the waterlogging-tolerant and sensitive genotypes. Consequently, the seedling parameters and waterlogging tolerance coefficients are efficient and reliable measurements for distinguishing the genotypes that are tolerant or sensitive to waterlogging stress.

## Figures and Tables

**Figure 1 plants-10-02294-f001:**
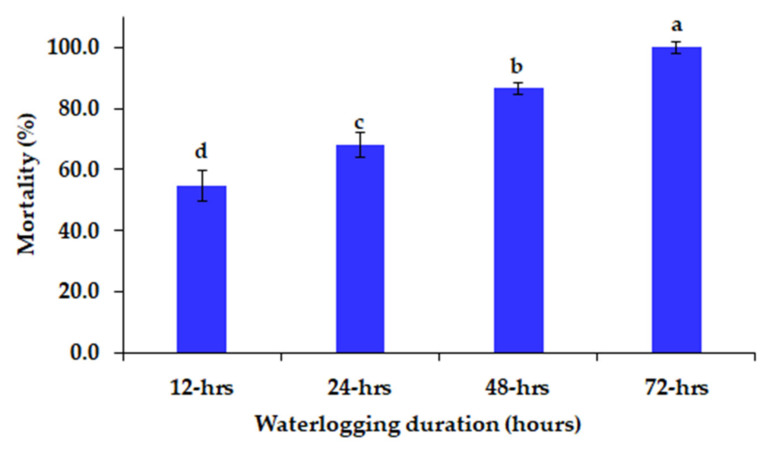
Mortality percentages of different sesame genotypes at the seedling stage under different durations of waterlogged conditions. The bars on the columns represent the SE, and different letters differ significantly by LSD (*p* < 0.01).

**Figure 2 plants-10-02294-f002:**
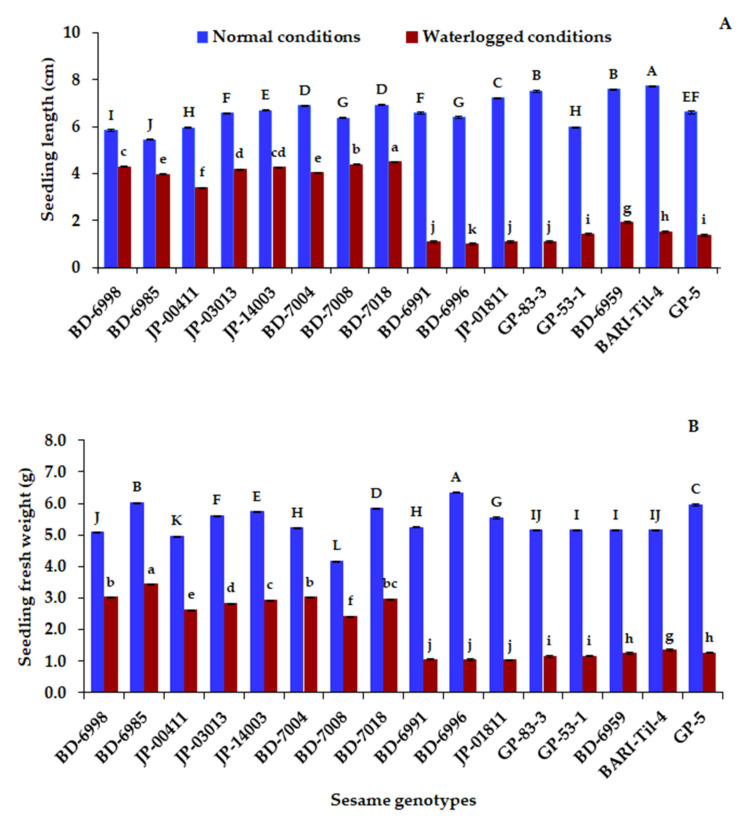
Seedling lengths (**A**) and fresh weights (**B**) of sixteen sesame genotypes under 48 h of waterlogging. Different uppercase letters indicate significant differences by LSD (*p* < 0.01) under normal conditions, while lowercase letters indicate waterlogged conditions.

**Figure 3 plants-10-02294-f003:**
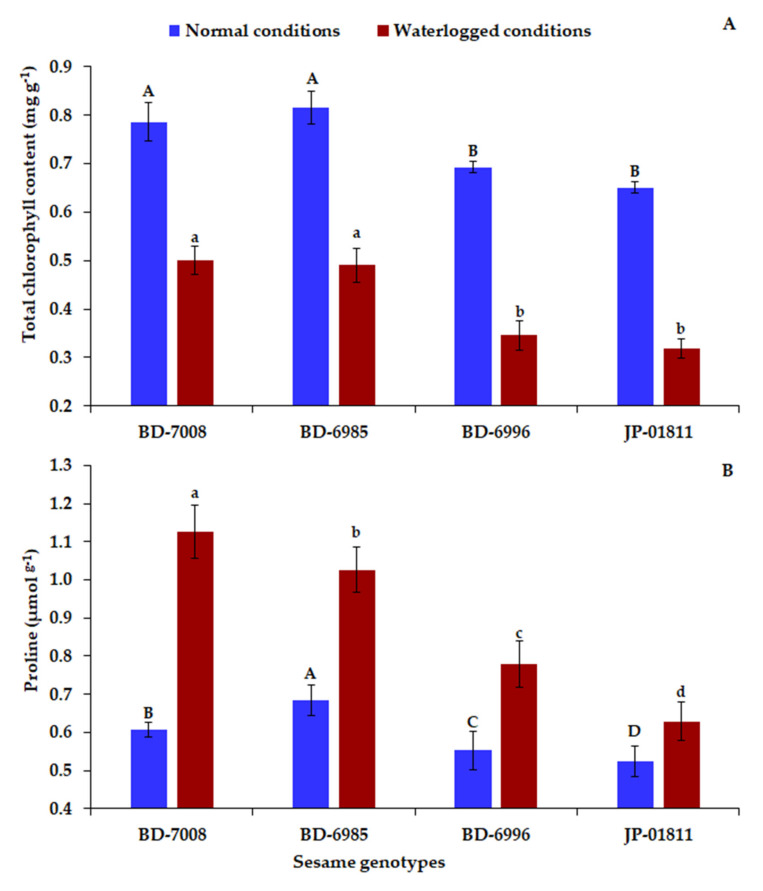
Total chlorophyll (**A**) and proline (**B**) contents of the four sesame genotypes at 48 h of waterlogging treatment. The bars on the columns correspond to SE and different uppercase letters indicate significant differences by LSD (*p* < 0.01) under normal conditions, while lowercase letters indicate waterlogged conditions.

**Figure 4 plants-10-02294-f004:**
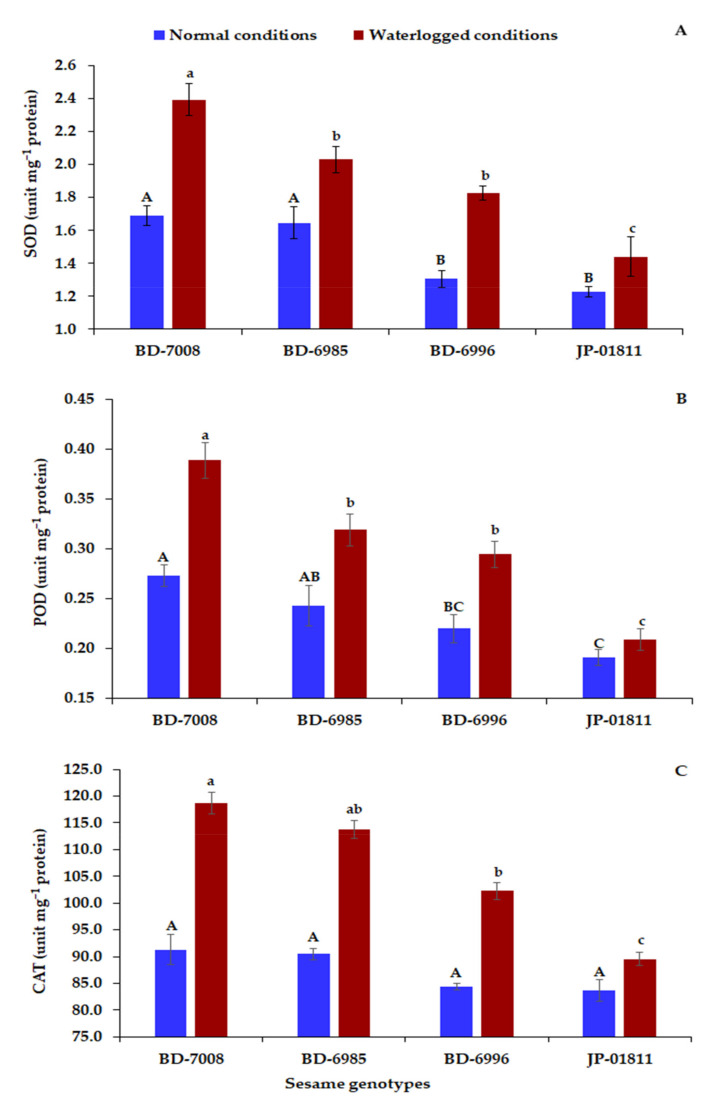
Impact of 48 h of waterlogged conditions on the activities of superoxide dismutase activity (SOD) (**A**), peroxidase (POD) (**B**), catalase (CAT) (**C**), and of four sesame genotypes. The bars on the columns correspond to SE and different uppercase letters indicate significant differences by LSD (*p* < 0.01) under normal conditions, while lowercase letters indicate waterlogged conditions.

**Figure 5 plants-10-02294-f005:**
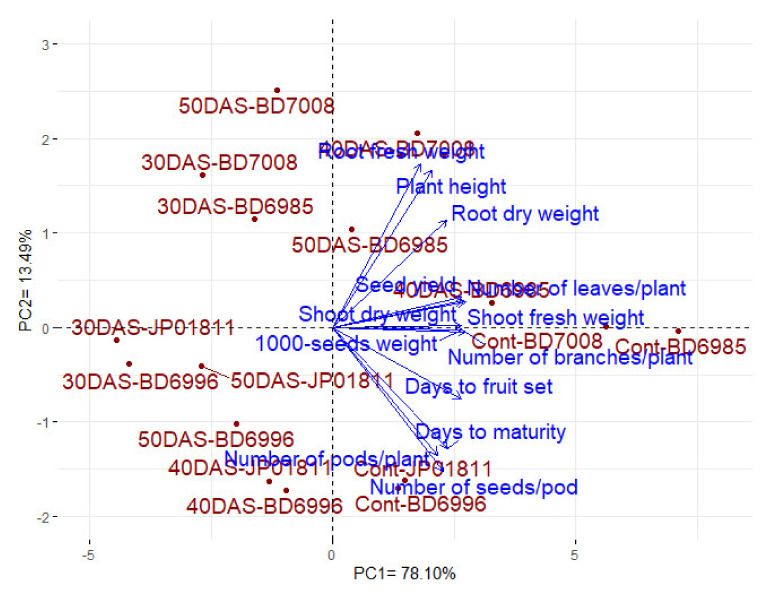
PCA biplot for the evaluated traits of four sesame genotypes (BD-7008, BD-6985, BD-6996, and JP-01811) under four waterlogging treatments—control, 30, 40, and 50 days after sowing, which are presented as Cont, 30D, 40D, and 50D.

**Table 1 plants-10-02294-t001:** List of the evaluated sesame genotypes.

No.	Accession Name	No.	Accession Name	No.	Accession Name	No.	Accession Name
1	BD-6971	31	JP-14017	61	KMR-100	91	JP-5
2	BD-6960	32	BD-7014	62	BD-7004	92	BD-6966
3	JP-01514	33	BD-6987	63	BD-6981	93	BD-6995
4	BD-6979	34	JP-35-2	64	BD-6978	94	AMA-102
5	BD-10167	35	BD-6962	65	JP-01411	95	BD-7001
6	BARI-Til-3	36	TT-152	66	MTR-66	96	JP-00515
7	BD-7015	37	BD-7023	67	BD-6984	97	BD-7017
8	BD-6986	38	BD-7006	68	BD-7000	98	KASI-22
9	BD-7027	39	BD-6972	69	AHM-150	99	BD-7029
10	BD-6980	40	BD-10166	70	BD-6959	100	BD-7013
11	BD-6990	41	IAH-333	71	JP-14004	101	JP-160
12	BD-6964	42	BD-7009	72	AC-316	102	BD-6999
13	BD-6992	43	BD-6982	73	BD-7026	103	GP-5
14	JP-14013	44	BD-6993	74	BD-7018	104	GP-21
15	BARI-Til-4	45	BD-6983	75	BD-6968	105	GP-35-1
16	BD-6985	46	BD-7022	76	BD-6980	106	GP-35-2
17	JP-02513	47	JP-14016	77	BD-6996	107	GP-83-1
18	BD-7012	48	JP-00411	78	JP-01311	108	GP-83-3
19	BD-6961	49	TRMR-55	79	BD-7003	109	GP-160
20	AHI-12	50	JP-01611-2	80	BD-6991	110	GP-181-1
21	BD-7019	51	JP-01711	81	BD-6998	111	GP-254
22	JP-10311	52	RISA-39	82	JP-01811	112	GP-694
23	BD-6974	53	BD-10165	83	KAI-112	113	GP-964
24	NRI-13	54	JP-03013	84	BD-6994	114	GP-212
25	BD-10164	55	BD-7021	85	BD-6989	115	GP-00411
26	BD-7011	56	MRI-31	86	JP-83-3	116	GP-515
27	JP-35-1	57	BD-7008	87	BD-7020	117	GP-01311
28	BD-6988	58	JP-14003	88	NQR-37	118	GP-01411
29	BD-7005	59	BD-6970	89	BD-7007	119	GP1514
30	S. Local	60	BD-6997	90	BD-7016		

**Table 2 plants-10-02294-t002:** Physico-chemical characterizations of the pot soil mixture used in the experiment.

Physical Properties		Chemical Properties	
Soil Characteristics	Value	Soil Characteristics	Value
Sand (%)	17.60	Soil pH	5.59
Silt (%)	47.30	Total N (%)	0.14
Clay (%)	35.10	Organic C (%)	0.68
Textual class	Silty clay loam	C:N ratio	5.0
Bulk density (g/cm^3^)	1.40	Available P (ppm)	6.68
Particle density (g/cm^3^)	2.61	Exchangeable K (meq/100 g)	0.13
Porosity (%)	47.4	Available Sulphur (ppm)	13.00
		Zn (ppm)	1.00

**Table 3 plants-10-02294-t003:** List of sesame genotypes that survived under varying durations of waterlogged conditions at the seedling stage.

Waterlogging Duration	Survived Sesame Genotypes	Number
**12 h**	S. local, BARI-Til-3, BARI-Til-4, BD-6959, BD-6960, BD-6964, BD-6966, BD-6970, BD-6972, BD-6974, BD-6978, BD-6979, BD-6980, BD-6981, BD-6984, BD-6985, BD-6986, BD-6987, BD-6989, BD-6991, BD-6993, BD-6994, BD-6996, BD-6998, BD-6997, BD-7000, BD7001, BD-7003, BD-7004, BD 7005, BD-7007, BD-7008, BD 7011, BD-7012, BD-7013, BD-7018, BD-7020, BD-7021, BD-7026, JP-00411, JP-00515, JP-01611-2, JP-03013, JP-14003, JP-14004, JP-140016, GP-5, GP-21, GP-35-2, GP-53-1, GP-83-1, GP-83-3, GP-160, GP-694	**54**
**24 h**	S. local, BARI-Til-3, BARI-Til-4, BD-6959, BD-6970, BD-6978, BD-6980, BD-6981, BD-6984, BD-6989, BD-6991, BD-6996, BD-6997, BD-6998, BD-6994, BD-7000, BD-7003, BD-7004, BD-7008, BD-7012, BD-7018, BD-7021, BD-7026, JP-00411, JP- 01611-2, JP-01811, JP-14003, JP-14004, JP-140016, BD-6985, JP-03013, GP-5, GP-21, GP 35-2, GP-53-1, GP 83-1, GP 83-3, GP-160	**38**
**48 h**	BARI-Til-4, BD-6959, BD-6985, BD-6991, BD-6996, BD-6998, BD-7008, BD-7004, BD-7018, JP-01811, JP-00411, JP-03013, JP-14003, GP-5, GP-53-1, GP-83-3	**16**
**72 h**	-	**0**

**Table 4 plants-10-02294-t004:** Pattern of survivability percentages of sixteen sesame genotypes in waterlogged conditions at 12, 24, and 48 h.

No.	Genotypes	Survivability (%)		
		12 h	24 h	48 h
1	BD-6998	100	100	90
2	BD-6985	100	100	100
3	JP-00411	100	100	90
4	JP-03013	100	100	90
5	JP-14003	100	100	90
6	BD-7004	100	90	90
7	BD-7008	100	100	100
8	BD-7018	100	100	90
9	BD-6991	100	80	60
10	BD-6996	100	70	45
11	JP-01811	100	60	40
12	GP-83-3	100	70	50
13	GP-53-1	100	70	50
14	BD-6959	100	70	50
15	BARI Til 4	100	65	60
16	GP-5	100	65	50

**Table 5 plants-10-02294-t005:** Growth inhibition rates and waterlogging tolerance coefficients (WTCs) at the seedling stage of sixteen sesame genotypes under waterlogged conditions for 48 h.

No.	Genotype	Inhibition% of Seedling Length	Inhibition% of Seedling Weight	WTC (Length)	WTC (Fresh Weight)
1	BD-6998	32.66 ^j^	42.28 ^k^	0.67 ^c^	0.54 ^b^
2	BD-6985	28.03 ^l^	40.21 ^l^	0.72 ^a^	0.60 ^a^
3	JP-00411	42.87 ^g^	47.39 ^j^	0.56 ^k^	0.53 ^bc^
4	JP-03013	35.86 ^i^	49.20 ^i^	0.64 ^ef^	0.51 ^c^
5	JP-14003	35.69 ^i^	48.56 ^ij^	0.66 ^cd^	0.51 ^c^
6	BD-7004	40.95 ^h^	42.18 ^k^	0.59 ^g^	0.58 ^a^
7	BD-7008	30.86 ^k^	40.49 ^l^	0.69 ^b^	0.59 ^a^
8	BD-7018	35.15 ^i^	48.83 ^i^	0.65 ^de^	0.51 ^c^
9	BD-6991	84.60 ^b^	64.81 ^h^	0.15 ^o^	0.35 ^d^
10	BD-6996	85.39 ^b^	83.61 ^b^	0.15 ^o^	0.16 ^h^
11	JP-01811	87.56 ^a^	86.82 ^a^	0.15 ^o^	0.18 ^h^
12	GP-83-3	85.33 ^b^	81.71 ^c^	0.12 ^p^	0.23 ^fg^
13	GP-53-1	77.07 ^e^	77.65 ^e^	0.23 ^m^	0.22 ^fg^
14	BD-6959	75.15 ^f^	75.94 ^f^	0.25 ^l^	0.24 ^ef^
15	BARI-Til-4	80.27 ^c^	73.74 ^g^	0.20 ^n^	0.26 ^e^
16	GP-5	79.08 ^d^	78.98 ^d^	0.20 ^n^	0.21 ^g^

Means followed by the different letters are significantly different by the LSD at *p* < 0.01.

**Table 6 plants-10-02294-t006:** Growth parameters of the sesame genotypes as influenced by waterlogging stress at different growth stages.

Studied Factors		PH	NL/P	NB/P	SFW	SDW	RFW	RDW
**Genotype (G)**								
BD-7008		93.25 ^b^	20.91 ^b^	8.50 ^a^	20.01 ^a^	2.65 ^b^	3.55 ^a^	0.74 ^b^
BD-6985		98.66 ^a^	23.67 ^a^	8.91 ^a^	20.54 ^a^	2.85 ^a^	3.00 ^b^	0.79 ^a^
BD-6996		68.33 ^c^	18.41 ^c^	6.91^b^	17.47 ^b^	2.45 ^c^	2.51 ^c^	0.60 ^c^
JP-01811		70.75 ^c^	18.33 ^c^	7.25 ^b^	16.98 ^b^	2.39 ^c^	2.49 ^c^	0.58 ^c^
*p*-value (G)		<0.001	0.018	0.021	0.05	0.016	0.005	0.011
**Waterlogging growth stage** **(W)**								
Control		88.17 ^a^	23.67 ^a^	10.16 ^a^	21.80 ^a^	3.00 ^a^	3.17 ^a^	0.76 ^a^
30 DAS		76.75 ^d^	16.83 ^d^	6.00 ^c^	15.77 ^d^	2.37 ^c^	2.58 ^c^	0.58 ^c^
40 DAS		85.17 ^b^	21.16 ^b^	8.41 ^b^	19.64 ^b^	2.61 ^b^	2.81 ^b^	0.70 ^ab^
50 DAS		80.92 ^c^	19.67 ^c^	7.00 ^c^	17.77 ^c^	2.36 ^c^	2.99 ^ab^	0.68 ^b^
*p*-value (W)		<0.001	<0.001	<0.001	<0.001	0.011	<0.001	0.002
**Interaction (G×W)**								
BD-7008	Control	97.00 ^a^	24.33 ^a^	11.00 ^a^	23.40 ^a^	3.18 ^a^	3.80 ^a^	0.81 ^a^
	30 DAS	90.00 ^b^	17.66 ^c^	6.33 ^c^	16.76 ^d^	2.32 ^c^	3.18 ^c^	0.56 ^c^
	40 DAS	95.33 ^a^	21.66 ^b^	9.33 ^b^	21.40 ^b^	2.77 ^b^	3.46 ^bc^	0.88 ^a^
	50 DAS	90.67 ^b^	20.00 ^b^	7.33 ^c^	18.50 ^c^	2.33 ^c^	3.75 ^ab^	0.71 ^b^
BD-6985	Control	104.67 ^a^	28.00 ^a^	11.66 ^a^	24.30 ^a^	3.49 ^a^	3.49 ^a^	0.93 ^a^
	30 DAS	91.33 ^c^	19.00 ^d^	6.66 ^c^	17.10 ^c^	2.54 ^c^	2.53 ^c^	0.69 ^d^
	40 DAS	102.00 ^a^	25.33 ^b^	9.66 ^b^	21.93 ^b^	2.81 ^b^	2.90 ^b^	0.80 ^b^
	50 DAS	96.67 ^b^	22.33 ^c^	7.66 ^c^	18.83 ^c^	2.55 ^c^	3.06 ^b^	0.74 ^c^
BD-6996	Control	72.33 ^a^	21.33 ^a^	8.66 ^a^	20.50 ^a^	2.69 ^a^	2.75 ^a^	0.65 ^a^
	30 DAS	64.67 ^c^	15.66 ^c^	5.33 ^d^	14.80 ^d^	2.32 ^c^	2.33 ^c^	0.51 ^b^
	40 DAS	69.00 ^b^	18.66 ^b^	7.33 ^bc^	17.76 ^b^	2.44 ^b^	2.44 ^bc^	0.61^ab^
	50 DAS	67.33 ^b^	18.00 ^b^	6.33 ^cd^	16.83 ^c^	2.35 ^c^	2.54 ^b^	0.64 ^a^
JP-01811	Control	78.67 ^a^	21.00 ^a^	9.33 ^a^	19.03 ^a^	2.64 ^a^	2.64 ^a^	0.64 ^a^
	30 DAS	61.00 ^d^	15.00 ^c^	5.66 ^d^	14.40 ^d^	2.31 ^bc^	2.27 ^c^	0.56 ^b^
	40 DAS	74.33 ^b^	19.00 ^ab^	7.33 ^bc^	17.46 ^b^	2.42 ^b^	2.44 ^bc^	0.51 ^b^
	50 DAS	69.00 ^c^	18.33 ^b^	6.66 ^cd^	16.93 ^c^	2.20 ^c^	2.61 ^ab^	0.61 ^ab^
*p*-value (G×W)		<0.001	0.001	0.02	0.03	0.05	0.004	0.001

PH is plant height (cm); NL/P is the number of leaves per plant; NB/P is the number of branches per plant; SFW is shoot fresh weight (g); SDW is shoot dry weight (g); RFW is root fresh weight (g); and RDW is root dry weight (g). Means followed by different letters under the same trait indicate significant differences by LSD (*p* < 0.01).

**Table 7 plants-10-02294-t007:** Yield-contributing traits of sesame genotypes as influenced by waterlogging stress at different growth stages.

Studied Factors		NP/P	NS/P	TGW	DFS	DM	SY
**Genotype (G)**							
BD-7008		35.75 ^a^	66.83 ^c^	8.66 ^a^	57.41 ^b^	76.66 ^c^	0.87 ^b^
BD-6985		35.66 ^a^	68.83 ^a^	9.08 ^a^	58.33 ^a^	78.41 ^a^	0.96 ^a^
BD-6996		35.75 ^a^	68.17 ^ab^	7.41 ^b^	57.08 ^bc^	77.58 ^b^	0.72 ^c^
JP-01811		35.50 ^a^	67.75 ^bc^	7.66 ^b^	56.91 ^c^	76.91 ^bc^	0.70 ^c^
*p*-value		0.048	0.021	0.031	0.008	0.004	0.001
**Waterlogging growth stage (W)**							
Control		37.08 ^a^	71.08 ^a^	10.16 ^a^	59.75 ^a^	79.75 ^a^	1.01 ^a^
30 DAS		34.08 ^c^	65.33 ^d^	7.08 ^c^	55.66 ^d^	75.33 ^c^	0.68 ^c^
40 DAS		36.08 ^ab^	68.33 ^b^	8.41 ^b^	57.91 ^b^	77.83 ^b^	0.82 ^b^
50 DAS		35.16 ^bc^	66.83 ^c^	7.20 ^c^	56.41 ^c^	76.66 ^bc^	0.73 ^b^
*p*-value		0.035	0.005	0.022	0.019	0.014	0.001
**Interaction (G×W)**							
BD-7008	Control	37.33 ^a^	70.66 ^a^	11.00 ^a^	60.66 ^a^	80.66 ^a^	1.13 ^a^
	30 DAS	34.66 ^b^	65.00 ^b^	7.00 ^c^	55.66 ^c^	74.33 ^c^	0.69 ^c^
	40 DAS	35.33 ^b^	66.00 ^b^	9.30 ^b^	58.00 ^b^	76.66 ^b^	0.86 ^b^
	50 DAS	34.66 ^b^	65.66 ^b^	7.33 ^c^	55.33 ^c^	75.00 ^c^	0.81 ^b^
BD-6985	Control	37.00 ^a^	72.66 ^a^	11.66 ^a^	61.33 ^a^	81.00 ^a^	1.11 ^a^
	30 DAS	33.66 ^c^	66.00 ^c^	7.33 ^c^	56.00 ^d^	76.66 ^c^	0.85 ^c^
	40 DAS	36.66 ^a^	69.66 ^b^	9.66 ^b^	58.66 ^b^	78.33 ^b^	1.00 ^b^
	50 DAS	35.33 ^b^	67.00 ^bc^	7.66 ^c^	57.33 ^c^	77.66 ^b^	0.86 ^c^
BD-6996	Control	36.66 ^a^	70.66 ^a^	8.66 ^a^	58.66 ^a^	79.00 ^a^	0.86 ^a^
	30 DAS	34.66 ^b^	65.00 ^c^	6.66 ^b^	55.33 ^c^	75.33 ^c^	0.64 ^d^
	40 DAS	36.00 ^a^	69.00 ^b^	7.33 ^b^	57.66 ^ab^	78.66 ^a^	0.72 ^bc^
	50 DAS	35.66 ^ab^	68.00 ^b^	7.0 ^b^	56.66 ^b^	77.33 ^b^	0.67 ^c^
JP-01811	Control	37.33 ^a^	70.33 ^a^	9.33 ^a^	58.33 ^a^	78.33 ^a^	0.94 ^a^
	30 DAS	33.33 ^c^	65.33 ^c^	7.33 ^b^	55.66 ^c^	75.00 ^c^	0.57 ^c^
	40 DAS	36.33 ^a^	68.66 ^b^	7.33 ^b^	57.33 ^ab^	77.66 ^ab^	0.71 ^b^
	50 DAS	35.00 ^b^	66.66 ^c^	6.66 ^b^	56.33 ^bc^	76.66 ^b^	0.58 ^c^
*p*-value		0.018	0.041	0.021	0.037	0.011	0.048

NP/P is the number of pods per plant; NS/P is the number of seeds per pod; TGW is the 1000-seed weight (g); DFS is days to fruit set (days); DM is days to maturity (days); and SY is seed yield (g/plant). Means followed by different letters under the same trait indicate significant differences by LSD (*p* < 0.01).

## Data Availability

The data presented in this study are available from the corresponding author upon request.

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
