# Peer review of "Assessing the Response of Diverse Sesame Genotypes to Waterlogging Durations at Different Plant Growth Stages"

_plants, 2021, doi:10.3390/plants10112294_

Round 1
Reviewer 1 Report
Dear Editor and authors, the ms plants-1362590 had as objetive to assess diverse sesame genotypes under various durations of waterlogging. All my comments and suggestions can be found in the attached file. After these changes, the ms can be recommended for publication in Plants.

Author Response
We would like to thank the reviewer for his time dedicated to our manuscript. All comments in the PDF file have been addressed in the revised version using track changes. Following the reviewer's suggestion, the data of Figures 3 and 4 have been reanalyzed to compare among the genotypes under normal conditions separated from waterlogging conditions as Figures 1 and 2. Moreover, the PCA biplot (Figure 5) has been reconstructed to present the genotype name instead of codes G1-G4. The conclusion has been revised and rewritten.
Reviewer 2 Report
The subject of the manuscript is consistent with the scope of the Journal.
The abstract faithfully conveys the scope of investigations and conclusions drawn. The keywords correspond well to the scope of the research.
The aim of this study was to evaluate diverse sesame genotypes under various durations of waterlogging to identify useful tolerant genetic resources and to investigate the response of sesame genotypes to waterlogging conditions at different ages to identify critical durations of waterlogging that negatively impact on sesame growth and productivity.
I think the paper needs some corrections:
1) add detailed information about experiment, e.g. plant harvest vegetation phase, etc.,
2) how much soil was added to each pot?
3) add the basic properties of the soil (type, physico-chemical properties, etc.) on which the pot experiment is established,
4) why LSD was resolved for significant data at P ≤ 0.05? LSD for laboratory and pot experiment should be at P ≤ 0.01, calculate the LSD for P ≤ 0.01,
5) standardize References section.
Paper needs some editorial corrections (see: Instructions for Authors).
You must check your paper very exactly and correct all mistakes and complete lacking data of papers.
Author Response
The subject of the manuscript is consistent with the scope of the Journal. The abstract faithfully conveys the scope of investigations and conclusions drawn. The keywords correspond well to the scope of the research. The aim of this study was to evaluate diverse sesame genotypes under various durations of waterlogging to identify useful tolerant genetic resources and to investigate the response of sesame genotypes to waterlogging conditions at different ages to identify critical durations of waterlogging that negatively impact sesame growth and productivity.
Re: We would like to thank the Reviewer for his time dedicated to our manuscript and presenting positive aspects in our manuscript.
I think the paper needs some corrections:
1) add detailed information about the experiment, e.g. plant harvest phase, etc.
Re: More details have been added, please see lines 166 and 169.
2) how much soil was added to each pot?
Re: The required details have been added in line 119.
3) add the basic properties of the soil (type, physico-chemical properties, etc.) on which the pot experiment is established.
Re: Soil physico-chemical properties have been added in Table 2.
4) why LSD was resolved for significant data at P ≤ 0.05? LSD for laboratory and pot experiment should be at P ≤ 0.01, calculate the LSD for P ≤ 0.01.
Re: All the data have been reanalyzed and LSD at P <0.01 has been used to separate the genotypes.
5) standardize References section
Re: The reference list has been standardized using Endnote.
The paper needs some editorial corrections (see: Instructions for Authors).
Re: The manuscript has been revised to follow the journal instructions
You must check your paper very exactly and correct all mistakes and complete lacking data of papers.
Re: The manuscript has been revised following the suggestions of the Reviewers
Reviewer 3 Report
The manuscript entitled: “Assessing Response of Diverse Sesame Genotypes to Waterlogging Duration at Different Plant Ages” provides some new insights about waterlogging and sesame genotypes. Authors used different waterlogging conditions of 12,24,48 and 72 hours and found two genotypes with the highest tolerance and two the most sensitive ones. Furthermore, they continue the experiment for those 4 genotypes at different ages and measured chlorophyll, proline content and Enzymatic Antioxidants Activity which gave new insights which can be used for future usage of the genotypes.
From my personal view, the experiments are well designed and the manuscript is well written. I do not have any suggested changes and I think that it is suitable for publication.
Author Response
We would like to thank the reviewer for the positive comment regarding our work.
Round 2
Reviewer 2 Report
The authors corrected all reviewer comments. Thank
you.
Author Response
We would like to thank the Reviewer for his time dedicated to reviewing our manuscript and his comments that helped us to improve the manuscript.